# The Effects of Different Forms of Butyric Acid on the Performance of Turkeys, Carcass Quality, Incidence of Footpad Dermatitis and Economic Efficiency

**DOI:** 10.3390/ani12111458

**Published:** 2022-06-04

**Authors:** Zbigniew Makowski, Krzysztof Lipiński, Magdalena Mazur-Kuśnirek

**Affiliations:** Department of Animal Nutrition and Feed Science, University of Warmia and Mazury in Olsztyn, 10-718 Olsztyn, Poland; zbyszek-tu@o2.pl (Z.M.); krzysztof.lipinski@uwm.edu.pl (K.L.)

**Keywords:** butyric acid glycerides, coated sodium butyrate, sodium butyrate, organic acids, poultry, performance, carcass quality, meat quality, FPD

## Abstract

**Simple Summary:**

Butyric acid is a short-chain organic acid with established antimicrobial properties. It decreases the pH of intestinal digesta and reduces the abundance of pathogenic bacteria, thus indirectly improving the growth performance of birds. In the present study, turkey diets were supplemented with different forms of butyric acid. The efficiency of bird production and carcass dressing percentage were improved when butyric acid glycerides or coated sodium butyrate were added to the diet. An improvement in footpad condition and an increase in the dry matter content of faeces were noted in birds fed experimental diets. The addition of butyric acid in various forms to turkey diets improved the economic efficiency of production. The results of this study suggest that different forms of butyric acid improve production efficiency, carcass traits, and footpad condition in turkeys. Therefore, sodium butyrate, coated sodium butyrate, and butyric acid glycerides can be valuable feed additives in turkey nutrition.

**Abstract:**

The aim of this study was to compare the efficacy of butyric acid glycerides (BAG), sodium butyrate (SB) and coated sodium butyrate (CSB) in turkey nutrition based on the growth performance of birds, carcass yield, meat quality, the dry matter (DM) content of faeces, the incidence of footpad dermatitis (FPD), and economic efficiency. A 105-day experiment was conducted on 400 BIG 6 female turkeys (4 treatments, 5 replications, 20 birds per replication). The addition of CSB and BAG to turkey diets improved the feed conversion ratio (FCR, *p* ≤ 0.05) and increased the values of the European Efficiency Index (EEI, *p* ≤ 0.01). The analysed forms of BA in turkey diets increased the concentration of DM in faeces (*p* ≤ 0.01) and decreased FPD incidence (*p* ≤ 0.01), which may suggest that all forms of butyrate improved litter quality and inhibit the risk for diarrhoea. The results of this study indicate that all forms of butyric acid can be valuable feed additives in turkey nutrition.

## 1. Introduction

Antibiotic growth promoters (AGPs) have been used for decades to improve the growth performance of livestock kept under adverse environmental conditions [1]. They were administered to animals to regulate intestinal microbiota by suppressing the growth of pathogenic microorganisms and toxin production [2,3]. However, the use of AGPs contributed to the antimicrobial resistance of bacteria such as *Salmonella*, *Campylobacter*, *Enterococcus* and *Escherichia coli* strains, thus posing a serious threat to animals and human health [2,4]. On 1 January 2006, the European Commission banned the use of antibiotics as growth promoters in animal nutrition in the European Union [5]. One of the solutions was the use of alternative growth promoters to maintain livestock health and improve performance. Probiotics, prebiotics, enzymes, phytobiotics, and organic acids have gained particular popularity in animal production [5,6].

Organic acids, such as acetic, butyric, propionic, fumaric, and lactic acids, exert beneficial effects on the gut health and growth performance of birds [7,8,9,10]. They can be incorporated into poultry diets as sodium, potassium, and calcium salts, in free form, as protected/coated salts, and as glycerides. Due to their antimicrobial properties, organic acids affect the gastrointestinal tract of birds by decreasing the pH of intestinal digesta [11], reducing the abundance of pathogenic bacteria, promoting the growth of *Lactobacillus* [12,13,14], improving nutrient digestibility [15,16], and enhancing the proliferation of intestinal epithelial cells [17,18].

Butyric acid (BA) is a SCFA that exhibits strong antimicrobial activity. It can be a rich source of energy for epithelial cells in the large intestine and the terminal segment of the small intestine, thus improving nutrient digestibility [19,20,21]. The butyric acid is commonly used in its butyrate form (calcium or sodium salt), because it is odourless [22]. Researchers have demonstrated that butyric acid is mainly absorbed in the upper digestive tract such as the crop, which makes it a less effective feed additive [23].

The mode of action is that once sodium butyrate reaches the bird stomach, it releases the Na ion. Due to the low pH, the other fraction -butyrate is rapidly converted to the undissociated form, defined as the butyric acid [21]. This form has antimicrobial properties and it can lower the gastric pH, increasing the conversion of pepsinogen to pepsin, which enhances the absorption rate of nutrients [24]. Butyric acid also stimulates the development of GALT (gut-associated lymphoid tissue) and the functional development of the gastrointestinal tract in terms of digestion and absorption of nutrients [25]. In addition, dietary BA may be attributed to the improvement in intestinal integrity. This acid decreases bacterial colonisation of the intestinal wall because it decreases pH in the digestive tract [26], resulting in less predisposition to diarrhoea [27].

However, butyrate efficacy has been demonstrated to increase when it is fed in a protected form such as encapsulation or by esterification of BA with glyceride [28]. A number of reports have documented that the supplementation of butyric acid had a positive effect on the performance and carcass composition of broiler chickens [11,29,30]. Accordingly, several experiments have demonstrated that also administering protected forms of BA results in enhanced animal performance and carcass yield [18,31]. The incomplete literature data available have demonstrated that the effectiveness of butyrate additive may depend on its forms [28]. However, there are no published studies investigating the efficiency of different forms of BA in turkey nutrition.

The research hypothesis postulates that the supplementation of turkey diets with different forms of BA can improve the growth performance of birds, carcass traits, the concentration of dry matter (DM) in faeces, the incidence of footpad dermatitis (FPD), and economic efficiency, and that the efficacy of BA depends on its form (salts, coated salts, or glycerides). Therefore, the aim of this study is to compare the efficacy of butyric acid glycerides (BAG), sodium butyrate (SB), and coated sodium butyrate (CSB) in turkey nutrition based on the growth performance of birds, carcass yield, meat quality, the DM content of faeces, FPD incidence, and economic efficiency.

## 2. Materials and Methods

### 2.1. Animals, Experimental Design and Diets

The experiment was conducted on 400 BIG-6 turkeys (female), which were randomly divided into four groups with five replicates and 20 birds per replicate. The birds were reared for 15 weeks from the first day of their life. Animals were housed on litter, in standard environmental conditions. For the nutrition of turkeys, standard, mash mixtures (nutritional value was consistent to the requirements of intensively growing birds) were used (Table 1). The animals had ad libitum access to water and feed. The nutrient content was determined by the Weende method [32]. The amount of metabolizable energy (ME_N_) and the content of minerals and amino acids in the diets were calculated according to the Nutrient Requirements of Poultry [33]. The diets contained mineral–vitamin premixes involving enzyme preparations (xylanase and phytase) and coccidiostat Clinacox (Starter 1,2 and Grower 1).

The birds from the control group (1) received a diet without feed additives. In experimental groups 2, 3, and 4, BA was added to turkey diets in the following forms and amounts:

Group 2 (SB)—sodium butyrate, n–butyric acid 98% ± 2% (Adimix CPS; Nutriad)—1 kg/t;

Group 3 (CSB)—coated sodium butyrate, coated n–butyric acid 30% ± 2% (Adimix 30 Coated; Nutriad)—3.3 kg/t;

Group 4 (BAG)—butyric acid glycerides, n–butyric acid 23–26% (C4 powder Monobutyrin; Silo S.r.l.)—3.4 kg/t.

Butyrate concentration was equal in all groups (approx. 780 g/t).

The body weight of turkeys and feed intake was determined after 8 and 15 weeks of life. Production parameters, such as the feed conversion ratio (FCR), daily cumulative mortality rate [%] and the European Efficiency Index (EEI), were calculated according to the following formulas:

FCR = (feed consumption (kg))/(BW gain (kg));

Daily cumulative mortality rate (%) = (number of dead turkeys × 100)/(number of turkeys in the experiment);

EEI = (liveability (%) × BW (kg) × 100)/(age (days) × FCR (kg)).

### 2.2. Sample Collection and Laboratory Analyses

On weeks 3, 6, 9, and 15, footpad dermatitis (FPD) scores were evaluated in 10 turkeys from each pen by visually inspecting and scoring following the classification described by Ekstrand et al. [34]. The 3-point scale scoring was performed based on the severity of the lesions in the footpad, where 0 indicated healthy birds without lesions and 2 appeared as the most severe ulcers, or swollen footpads. Footpad lesions were scored and were calculated as follows: 

The FPD score = [(number of feet in class 1 × 0.5 + number of feet in class 2 × 2)/sample size] × 100.

At 6 and 15 weeks of age, 15 birds (3 per replicate) representing the average body weight of each treatment were selected and killed. Carcass characteristics (dressing percentage and breast muscles) were determined. The percentage content of gizzard, liver, and heart in live body weight was determined. A total of 50 g samples of fresh breast muscles were assayed for proximate chemical composition [32].

Faeces were collected for DM analyses in weeks 3, 6, 9, and 15 of the experiment. Deep-frozen samples were lyophilised at −70 °C (lyophiliser Alpha 1–2 LD plus, CHRIST, Osterode am Harz, Germany). To determine DM content, faecal samples were dried in the BINDER ED 720 dryer with forced air circulation, at a temperature of 105 °C, to constant weight.

A simplified analysis of the economic efficiency of turkey production was performed based on the final BW of birds, feed intake, and the average price of diets in February 2022. The applied feed additives, SB (1.0 kg/t), CSB (3.3 kg/t), and BAG (4.0 kg/t), increased feed cost by EUR 3.6, 12.54, and 15.2 per t, respectively. Total costs were calculated based on the assumption that feed cost accounted for 70% of total costs.

### 2.3. Statistical Analysis

Three various forms of BA were an experimental factor: sodium butyrate, coated sodium butyrate, and butyric acid glycerides. The results were processed statistically by the one-way ANOVA and Duncan’s test. The arithmetic mean, the standard error of the mean (SEM), and the level of significance (*p* ≤ 0.05) were calculated for all results. All calculations were performed in the Statistica 12.0 program (StatSoft, Kraków, Poland).

## 3. Results

### 3.1. Growth Performance

The growth performance parameters of turkeys are presented in Table 2. After 8 weeks of the experiment, the birds fed diets supplemented with various forms of BA were characterised by higher BW (*p* ≤ 0.05) and feed intake (*p* ≤ 0.01) compared with the control group. After 15 weeks, the analysed feed additives had no effect (*p* > 0.05) on the final BW of turkeys and feed intake.

After 8 weeks of experiment, FCR was similar in all groups (*p* > 0.05). After 15 weeks, birds fed diets supplemented with CSB and BAG were characterised by lower values of the FCR (*p* ≤ 0.05) as compared with the control group.

Birds fed diets supplemented with CSB or BAG were characterised by higher values of the EEI (*p* < 0.01) compared with birds from the other groups.

### 3.2. Carcass Quality and Breast Muscle Composition

The carcass quality parameters and the chemical composition of breast muscles of turkeys are presented in Table 3. The analysed additives had no effect (*p* > 0.05) on carcass dressing percentage or carcass tissue composition in turkeys aged 6 weeks. In week 15 of the experiment, turkeys receiving BAG diets were characterised by a higher (*p* < 0.05) dressing percentage than birds fed SB diets. The percentage content of breast muscles in the carcass was highest in turkeys fed diets supplemented with CSB, compared with control group birds (difference of approx. 5%; *p* < 0.05). The proportion of gizzard in the carcass was higher (by 14% on average, *p* < 0.05) in birds fed non-supplemented diets compared with birds receiving SB or CSB. Dietary supplementation with SB significantly decreased the proportion of liver in the carcass (*p* < 0.01).

The tested feed additives had no influence (*p* > 0.05) on the content of DM, crude ash, or crude protein in turkey meat. The crude fat content of breast muscles was lower in birds fed BAG-supplemented diets compared with the remaining groups (*p* ≤ 0.05).

### 3.3. Footpad Dermatitis and Faecal Dry Matter

Changes in the mean FPD score and the concentration of DM in faeces are presented in Table 4. A strong positive effect of the analysed forms of BA on the FPD score was noted in weeks 9 and 15 of the experiment (*p* < 0.01). Turkeys receiving BA in different forms were characterised by a higher concentration of DM in faeces (*p* < 0.01) than control group birds.

### 3.4. Economic Evaluation

A simplified economic evaluation (Table 5) revealed that at the weighted average price of the control diet (EUR 477 per t), feed cost per kg live weight was EUR 1.12 in all experimental groups. This resulted from the improved performance of birds receiving feed additives containing BA. Total costs, including feed cost and indirect costs, were lower by EUR 0.01 per kg live weight in group SB than in the control group and the remaining experimental groups. As a result, revenue per kg live weight was higher in group SB than in the other groups (EUR 0.04 vs. 0.03). The profit resulting from the addition of the tested feed additives to turkey diets reached EUR 0.099 per bird in group SB, and EUR 0.006 per bird in groups CSB and BAG. 

## 4. Discussion

Butyric acid is often added to poultry diets. However, the efficacy of BA or protected forms of BA in turkey nutrition remains insufficiently investigated. In the present study, birds fed diets supplemented with various forms of BA were characterised statistically by higher BW after 8 weeks of the experiment. However, feed supplementation with various forms of butyric acid after 15 weeks of experiment caused a slight increase in the body weight of turkeys, but it was not statistically significant. Available literature data indicate that both unprotected and protected forms of BA may be used as growth stimulators and may improve BW gain and FCR value in poultry [18,22,29,35,36,37,38]. 

Butyric acid has a direct effect on the proliferation, maturation, and differentiation of mucosal cell, because it can influence gene expression, protein synthesis, and finally, promotion of the growth performance and carcass yield of birds [35]. Coated butyrate is characterised by higher bioavailability in the small intestine of birds, which increases its efficacy and improves nutrient utilisation [28]. In turn, BA glycerides are less affected by gizzard pH and can reach the distal segments of the gastrointestinal tract where undissociated acids are released by intestinal lipases [36]. Released BA can directly influence gut morphology, microbiota, and digestion processes, thus improving feed conversion and carcass traits, which was confirmed in our study. Only coated butyric acid and glycerides contributed to decreasing FCR values and increasing EEI values in turkeys. In addition, turkeys fed BAG-supplemented diets were characterised by higher carcass quality and the addition of CSB to feed increased the proportion of breast muscles in the carcass. The stimulatory action of the increased carcass yield of protected butyrate-supplemented animals has been already described in other studies [31,37]. This result is similar to what was observed in birds supplemented with unprotected sodium butyrate, suggesting that both butyrate derivatives are effective in promoting meat production [11,29]. However, the breast muscles weight and dressing percentage were closely correlated with the final BW of animals, which was confirmed in our study. 

In our study, turkeys fed BAG-supplemented diets were characterised by a lower concentration of crude fat in breast muscle. Bedford et al. [30] noted increased breast muscle deposition and decreased abdominal fat deposition in broiler chickens receiving butyrate glyceride blends (mono- and triglycerides). This finding is consistent with the results obtained by Yin et al. [38], who found that the concentrations of abdominal and intramuscular fat decreased in broiler chickens fed diets supplemented with BA derivatives. They hypothesised that butyric acid inhibits lipolysis and lipogenesis pathways, which may lead to decreased fat deposition in carcasses or breast muscles [38,39].

The butyric acid has the ability to change from an undissociated to dissociated form (depending on the environmental pH), which enhances its antimicrobial effect. When the acid is in the undissociated form, it can diffuse through the semi-permeable membrane of the bacteria into the cell cytoplasm [40]. Inside the cell, where the pH is maintained near 7, the acid will dissociate and suppress nutrient transport systems and bacterial cell enzymes (e.g., catalases or decarboxylases) [1]. The efficacy of an acid in inhibiting harmful microorganism is dependent on its pKa value, which is the pH at which 50% of the acid is dissociated. BA has a pKa equal to a 4.81 value that dissociates in the crop [41]. The butyrate needs to be in an undissociated state before reaching the lower part of the small intestine to exert its stronger antimicrobial effect [42]. The protected forms of butyric acid may overcome this problem, as they are available even in proximal and distal portions of the birds’ intestine [43]. However, in the present study, all forms of BA added to turkey diets contributed to an increase in the DM content of faeces and a decrease in FPD incidence (9 and 15 weeks of the experiment). Due to its antimicrobial properties, BA decreases the pH of intestinal digesta, reduces the abundance of pathogenic bacteria (such as *Escherichia coli*, *Clostridium* spp., *Salmonella*), and promotes the growth of beneficial bacteria in the small intestine of birds [44,45,46,47]. Stable bacterial populations minimise the risk of enteritis and diarrhoea in poultry [48]. An increase in faecal DM points to reduced water excretion, which positively affects litter quality and reduces FPD incidence [49,50]. Improved FPD scores in birds receiving organic acids have also been reported by other authors [51,52,53]. Previous research has confirmed that organic acids had a beneficial influence on gut health in poultry [12,13,54,55], but the effect of BA on FPD scores has not been investigated to date.

In the present study, the profit resulting from the addition of the tested feed additives to turkey diets reached EUR 0.099 per bird (group SB) and EUR 0.006 per bird (groups CSB and BAG). Similar observations were made by other authors, who found that organic acids added to poultry diets improved economic efficiency [9,56,57]. Organic acids contribute to increasing the BW gain and final BW of birds or decreasing the FCR, which directly affects economic profit. However, the final economic results depended on the feed prices and live body weight of turkeys in an experimental period.

## 5. Conclusions

The addition of different forms of BA to turkey diets improved production efficiency. The analysed forms of BA in turkey diets increased the concentration of DM in faeces and decreased FPD incidence, which may suggest that all forms of butyrate improved litter quality and inhibit the risk for diarrhoea. All forms of BA can be valuable feed additives in turkey nutrition.

## Figures and Tables

**Table 1 animals-12-01458-t001:** The composition and nutritional value of turkey diets.

Specification	Starter 1	Starter 2	Grower 1	Grower 2	Finisher
	0–3 Weeks	4–6 Weeks	7–9 Weeks	10–12 Weeks	13–15 Weeks
Ingredient [g/kg]					
Wheat	261.9	310.3	416.7	515.8	590.4
Maize	200.0	200.0	150.0	100.0	100.0
Soybean meal	358.2	360.9	347.4	300.0	225.1
Full-fat soybeans	100.0	50.0	-	-	-
Blood meal	20.0	10.0	-	-	-
Soybean oil	5.2	19.2	39.1	45.1	47.8
L-lysine HCl	3.1	3.6	3.7	3.2	4.0
DL-methionine	3.5	2.6	2.4	2.6	2.5
L-threonine	0.7	0.7	1.1	0.7	1.0
Limestone	18.8	14.5	13.4	11.1	9.7
Calcium phosphate	22.1	19.9	17.7	13.1	11.1
Sodium bicarbonate	0.1	1.3	1.2	0.7	0.7
NaCl	2.0	1.9	2.2	2.6	2.7
Feed enzymes	0.1	0.1	0.1	0.1	0.1
Premix *	5.0	5.0	5.0	5.0	5.0
Nutritional value					
ME_N_, [kcal/kg]	2800	2880	3000	3100	3200
Crude Protein, %	27.51	25.49	23.24	22.31	20.13
Crude Fibre, %	3.22	2.96	2.90	3.19	3.22
Ether Extract, %	3.52	4.13	5.31	5.77	6.66
Lysine, [%]	1.77	1.65	1.45	1.30	1.17
Methionine + Cysteine, [%]	1.15	1.02	0.95	0.93	0.85
Ca, [g]	1.40	1.20	1.15	1.00	0.90
P available, [g]	0.70	0.65	0.60	0.50	0.45
Na, [g]	0.13	0.15	0.15	0.15	0.15

* Premix composition: Starter—12,500 IU vit. A, 4500 IU vit. D3, 87.5 mg vit. E, 3.75 mg vit. K3, 3.5 mg vit. B1, 10 mg vit. B2, 75 mg niacin, 22.5 mg pantothenic acid, 6.0 mg vit. B6, 30 µg vit. B12, 2.5 mg folic acid, 400 µg biotin, 800 mg choline chloride, 92.5 mg Fe, 130 mg Mn, 20 mg Cu, 105 mg Zn, 2.5 mg J, and 0.3 mg Se; Grower—11,500 IU vit. A, 4140 IU vit. D3, 80.5 mg vit. E, 3.45 mg vit. K3, 3.22 mg vit. B1, 9.2 mg vit. B2, 69 mg niacin, 20.7 mg pantothenic acid, and 5.52 mg vit. B6, 37.6 µg vit. B12, 2.3 mg folic acid, 368 µg biotin, 600 mg choline chloride, 85.1 mg Fe, 120 mg Mn, 18.4 mg Cu, 96.6 mg Zn, 2.3 mg J, and 0.26 mg Se; Finisher—10,500 IU vit. A, 3780 IU vit. D3, 66.5 mg vit. E, 2.85 mg vit. K3, 2.66 mg vit. B1, 7.6 mg vit. B2, 57 mg niacin, 17.1 mg pantothenic acid, and 4.6 mg vit. B6, 22.8 µg vit. B12, 1.9 mg folic acid, 304 µg biotin, 400 mg choline chloride, 70.3 mg Fe, 98.8 mg Mn, 15.2 mg Cu, 79.8 mg Zn, 1.9 mg J, and 0.23 mg Se.

**Table 2 animals-12-01458-t002:** Growth performance of turkeys.

Specification	Groups				SEM	*p*-Value
	1-Control	2-SB	3-CSB	4-BAG		
Body weight, g						
8 weeks	3.77 ^a^	3.91 ^b^	3.97 ^b^	3.87 ^b^	0.024	0.010
15 weeks	9.59	9.67	9.78	9.79	0.033	0.087
Feed intake-cumulative, g						
8 weeks	6.49 ^a^	6.75 ^b^	6.84 ^b^	6.68 ^b^	0.041	0.002
15 weeks	22.41	22.26	22.21	22.16	0.108	0.890
FCR-cumulative, kg/kg						
8 weeks	1.75	1.75	1.75	1.74	0.005	0.690
15 weeks	2.35 ^a^	2.32 ^ab^	2.29 ^b^	2.28 ^b^	0.010	0.015
Mortality, %	7.00	7.00	7.00	6.00	0.547	0.907
EEI, points	407.65 ^a^	417.19 ^a^	427.75 ^b^	429.87 ^b^	2.537	<0.001

Different superscripts in same row are significant or trending (a/b: *p* ≤ 0.05). SEM = standard error of the mean; FCR—feed conversion ratio; EEI—European Efficiency Index; Groups: 2-SB—sodium butyrate, n–butyric acid 98% ± 2%—1 kg/t, 3-CSB—coated sodium butyrate, coated n–butyric acid 30% ± 2%—3.3 kg/t, 4-BAG—butyric acid glycerides, and n–butyric acid 23–26%—3.4 kg/t.

**Table 3 animals-12-01458-t003:** Carcass quality characteristics and chemical composition of breast muscles in 15-week-old turkeys.

Specification	Groups				SEM	*p*-Value
	1-Control	2-SB	3-CSB	4-BAG		
Dressing percentage, %	81.56 ^ab^	81.21 ^b^	82.45 ^ab^	82.70 ^a^	0.242	0.087
Organ proportions in the carcass, %						
Breast muscles	26.78 ^b^	27.73 ^ab^	28.28 ^a^	28.08 ^ab^	0.238	0.015
Gizzard	1.15 ^a^	1.00 ^b^	0.98 ^b^	1.05 ^ab^	0.021	0.026
Heart	0.35	0.38	0.37	0.36	0.006	0.209
Liver	1.71 ^a^	0.96 ^b^	1.62 ^a^	1.49 ^a^	0.068	<0.001
Chemical composition of breast muscles, %						
Dry matter	26.25	25.55	26.29	26.22	0.1660	0.309
Crude ash	1.19	1.23	1.20	1.22	0.01	0.060
Crude protein	24.99	24.78	24.93	25.07	0.07	0.564
Crude fat	0.58 ^a^	0.66 ^a^	0.65 ^a^	0.40 ^b^	0.03	0.001

Different superscripts in same row are significant or trending (a/b: *p* ≤ 0.05). SEM = standard error of the mean; Groups: 2-SB—sodium butyrate, n–butyric acid 98% ± 2%—1 kg/t, 3-CSB—coated sodium butyrate, coated n–butyric acid 30% ± 2%—3.3 kg/t, 4-BAG—butyric acid glycerides, and n–butyric acid 23–26%—3.4 kg/t.

**Table 4 animals-12-01458-t004:** Footpad dermatitis scores.

Specification		Groups			SEM	*p*-Value
	1-Control	2-SB	3-CSB	4-BAG		
FPD score **						
−3 weeks	11.50	5.00	7.00	9.00	1.519	0.498
−6 weeks	15.50	10.50	9.00	10.00	1.919	0.656
−9 weeks	60.00 ^a^	44.50 ^b^	38.50 ^b^	42.00 ^b^	2.052	<0.001
−15 weeks	92.50 ^a^	78.50 ^b^	75.00 ^b^	74.00 ^b^	2.200	<0.001
Faecal dry matter, %						
−3 weeks	14.61 ^b^	17.83 ^a^	17.85 ^a^	19.48 ^a^	0.482	≤0.01
−6 weeks	12.88 ^b^	17.44 ^a^	16.63 ^a^	16.49 ^a^	0.469	≤0.01
−9 weeks	14.65 ^c^	20.23 ^a^	21.82 ^a^	18.14 ^b^	0.645	≤0.01
−15 weeks	12.71 ^b^	23.12 ^a^	23.80 ^a^	23.14 ^a^	1.135	<0.01

Different superscripts in same row are significant or trending (a/b/c: *p* ≤ 0.05). SEM = standard error of the mean; ** FPD score, 0–2 scale; Groups: 2-SB—sodium butyrate, n–butyric acid 98% ± 2%—1 kg/t, 3-CSB—coated sodium butyrate, coated n–butyric acid 30% ± 2%—3.3 kg/t, 4-BAG—butyric acid glycerides, and n–butyric acid 23–26%—3.4 kg/t.

**Table 5 animals-12-01458-t005:** Economic evaluation (per bird).

Specification	Groups			
	1-Control	2-SB	3-CSB	4-BAG
Price per kg of live weight, EUR/kg	1.63			
Cost of feed additives				
SB, EUR/kg	-	3.60	-	-
CSB, EUR/kg	-	-	3.80	-
BAG, EUR/kg	-	-	-	3.80
Average cost of diets, EUR/kg	0.477	0.481	0.490	0.493
		+0.004	+0.013	+0.015
Farm Performance:				
Live weight, kg	9.59	9.67	9.78	9.79
FCR, kg/kg	2.35	2.32	2.29	2.28
Feed consumed, kg	22.54	22.43	22.40	22.32
Financial Performance (EUR)*:*				
Feed price, kg	0.477	0.481	0.491	0.494
Feed cost/bird	10.76	10.79	10.97	11.00
Feed cost/kg live weight	1.12	1.12	1.12	1.12
Total cost/bird	15.37	15.42	15.68	15.71
Total cost/kg live weight	1.60	1.59	1.60	1.60
Revenue/kg	1.63	1.63	1.63	1.63
Revenue/bird	15.63	15.76	15.94	15.96
Margin/kg live weight	0.03	0.04	0.03	0.03
Margin/bird	0.288	0.387	0.293	0.294
Difference		0.099	0.006	0.006

Groups: 2-SB—sodium butyrate, n–butyric acid 98% ± 2%—1 kg/t, 3-CSB—coated sodium butyrate, coated n–butyric acid 30% ± 2%—3.3 kg/t, 4-BAG—butyric acid glycerides, and n–butyric acid 23–26%—3.4 kg/t.

## Data Availability

The datasets generated during and/or analyzed during the current study are available from the corresponding author on reasonable request.

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
