# Peer review of "The Effects of Different Forms of Butyric Acid on the Performance of Turkeys, Carcass Quality, Incidence of Footpad Dermatitis and Economic Efficiency"

_animals, 2022, doi:10.3390/ani12111458_

Round 1
Reviewer 1 Report
The aim of the study was to compare the efficacy of butyric acid glycerides (BAG), sodium butyrate (SB) and coated sodium butyrate (CSB) in turkey nutrition based on the growth performance of birds, carcass yield, meat quality, the dry matter (DM) content of feces, the incidence of footpad dermatitis (FPD), and economic efficiency. The subject of the manuscript falls within the general scope of the journal. The experiment has been well planed. The manuscript is interesting and brings some new information. In my opinion, it could be suitable for publication after revision, and after re-review.
Title
The economic efficiency was analyzed in the study. Therefore this information should be indicate in the title.
Abstract
Line 33: Double dot.
Introduction
Lines 38-41: Please add the references.
Lines 44-45 and throughout the manuscript: The Latin scientific names of all organisms should be italicized.
Lines 71-72: „The positive effects of BA on 71 productivity and carcass yield in poultry have been well documented ...”. Please indicate a new aspect of the study.
Materials and Methods
Line 87: “ad libitum” should be italicized.
Line 89: Please explain the abbreviation used the first time.
Table 1. Please explain the abbreviation used in the table.
Line 138: “25 g samples were collected...”. The content of dry matter, crude ash, crude protein, and crude fat was determined (I think in double repetitions). Did you have enough meat??
Line 152: Please indicate the experimental factor.
Line 152: In your experimental design, the best way for means comparison is the use of orthogonal contrasts. Please explain why you did not use this method.
Lines 153-154: “the level of significance (p<0.05) were calculated for all results”. Please explain why you write „p ≤ 0.05” in the next sections.
Results
Line 161 and throughout the manuscript: Please indicate the significance level when you write: „had no effect”, „ was similar”, „no influence”, etc.
Lines 170-172: “In weeks 6, 7 and 8 of the experiment, feed intake was higher in turkeys receiving various forms of BA than in control group birds (p ≤ 0.01).” It's not true. Please refer to the data in Table 3 for the 7th week and the 1st and 4th groups.
Lines 195-197: “Turkeys receiving diets with the addition of CSB and BAG were characterized by lower values of the FCR (p < 0.05) ... than birds from the other groups.” It's not true. Please refer to the data in Table 5 for the 2nd group.
Discussion
Lines 195-197: “Turkeys receiving diets with the addition of CSB and BAG were characterized by lower values of the FCR (p < 0.05) ... than birds from the other groups.” It's not true. Please refer to the data in Table 5 for the 2nd group.
Lines 195-197: I suggest to remove this sentence.
Lines 195-197: “Various forms of BA added to turkey diets in this study contributed to … a decrease in FPD incidence.”. Please refer to the data in Table 8 – only in weeks 9 I 15.
Author Response
The aim of the study was to compare the efficacy of butyric acid glycerides (BAG), sodium butyrate (SB) and coated sodium butyrate (CSB) in turkey nutrition based on the growth performance of birds, carcass yield, meat quality, the dry matter (DM) content of feces, the incidence of footpad dermatitis (FPD), and economic efficiency. The subject of the manuscript falls within the general scope of the journal. The experiment has been well planed. The manuscript is interesting and brings some new information. In my opinion, it could be suitable for publication after revision, and after re-review.
We would like to thank the Reviewer for a thorough perusal of the manuscript and valuable comments and suggestions which have enabled us to improve its quality.
Title
The economic efficiency was analyzed in the study. Therefore this information should be indicate in the title.
The information was added. L4
Abstract
Line 33: Double dot.
The relevant correction has been made
Introduction
Lines 38-41: Please add the references.
The missing information has been added. L41-43
Lines 44-45 and throughout the manuscript: The Latin scientific names of all organisms should be italicized.
The relevant corrections have been made
Lines 71-72: „The positive effects of BA on 71 productivity and carcass yield in poultry have been well documented ...”. Please indicate a new aspect of the study.
The missing information has been added. „However there are no published studies investigating the efficiency of different forms of BA in turkey nutrition.” L80
Materials and Methods
Line 87: “ad libitum” should be italicized.
Line 89: Please explain the abbreviation used the first time.
Table 1. Please explain the abbreviation used in the table.
The relevant corrections have been made L.101,103 Table 1
Line 138: “25 g samples were collected...”. The content of dry matter, crude ash, crude protein, and crude fat was determined (I think in double repetitions). Did you have enough meat??
The relevant correction has been made L. It was a mistake, in fact, in experiment were used 50 g of meat to analyze chemical composition.L162
Line 152: Please indicate the experimental factor.
The missing information has been added L.179
Line 152: In your experimental design, the best way for means comparison is the use of orthogonal contrasts. Please explain why you did not use this method.
In our study we compare three different sources of BA. In this case we regularly use a one-way ANOVA analysis with post-hoc test (Duncan) to confirm where the differences occurred between a few groups.
Lines 153-154: “the level of significance (p<0.05) were calculated for all results”. Please explain why you write „p ≤ 0.05” in the next sections.
It was a mistake, it should be “p ≤ 0.05” in throughout the manuscript. The relevant correction has been made L182
Results
Line 161 and throughout the manuscript: Please indicate the significance level when you write: „had no effect”, „ was similar”, „no influence”, etc.
The missing information has been added.
Lines 170-172: “In weeks 6, 7 and 8 of the experiment, feed intake was higher in turkeys receiving various forms of BA than in control group birds (p ≤ 0.01).” It's not true. Please refer to the data in Table 3 for the 7th week and the 1st and 4th groups.
Lines 195-197: “Turkeys receiving diets with the addition of CSB and BAG were characterized by lower values of the FCR (p < 0.05) ... than birds from the other groups.” It's not true. Please refer to the data in Table 5 for the 2nd group.
As suggested by Reviewer 2, the tables with growth performance were modified. Now, there is only one table (number 2), but as suggested by the Reviewer the relevant corrections have been made. The description of results was changed.
Discussion
Lines 195-197: “Turkeys receiving diets with the addition of CSB and BAG were characterized by lower values of the FCR (p < 0.05) ... than birds from the other groups.” It's not true. Please refer to the data in Table 5 for the 2nd group.
Lines 195-197: I suggest to remove this sentence.
Lines 195-197: “Various forms of BA added to turkey diets in this study contributed to … a decrease in FPD incidence.”. Please refer to the data in Table 8 – only in weeks 9 I 15.
The relevant correction has been made

Reviewer 2 Report
The manuscript describes the results of a trial designed to evaluate the effect of different forms of butyric acids as additives for turkeys. The trial appears to be well performed, however, the results deserved to be better described and discussed, especially to focus the additive forms comparison and likely links among variables. Please, see the following comments.
Introduction provides excessive general explanations about antibiotic alternatives, but the explanations about the availability and state of the art of different butyrate forms and justification of the hypothesis is scarce. Is there any study focusing the efficacy of butyrate forms in broilers? The study proposes to explore the effect in different variables (performance, carcass quality, meat composition, footpad dermatitis, dry matter content of feces), which are the antecedents and/or mechanisms that could justify the hypothesis.
Line115.- please, give information about the butyrate equivalent content in each treatment. Were they equal?
Table 2, Table 3, and Table 4 should be simplified considering the different dietary periods (starter 1, starter 2, Grower 1 and 2, and finisher diets) or even reducing the number of periods, instead of a week presentation. Table 5 is repeating values shown in other Tables. Merge the information. Please consider the possibility of moving from 4 Tables to only 1 for performance results.
Merge Table 6 and 7 , after deleting values from Week 6 (average information can be described in the body of the text.
Merge Table 8 and 9
Table 10 contain very interesting information, but it depends on prices which are continuously evolving (see for example, the extraordinary change on the prices from February 22 to April 22). This information doesn´t deserve to be published in detail.
Discussion should be clearly improved. In the present version, the discussion is organized to discuss all variables sequentially with the main objective of finding previous reports that are or not coincident with these results. At his respect, discussion is not ambitious and loss the opportunity to identify likely general mechanisms that can explain the results, even considering the discussion of results on a more transversal way. As there is not a clear hypothesis, discussion doesn´t attend a main objective to identify if butyrate exert effects on turkeys, and if these effects depends on the additive forms. When differences are observed derived from the form, a proper discussion is required.
Conclusions should not be a summary of results. For example, it is not evident if coating of glycerides promotes differences from sodium butyrate.
Author Response
The manuscript describes the results of a trial designed to evaluate the effect of different forms of butyric acids as additives for turkeys. The trial appears to be well performed, however, the results deserved to be better described and discussed, especially to focus the additive forms comparison and likely links among variables. Please, see the following comments.
We would like to thank the Reviewer for a thorough perusal of the manuscript and valuable comments and suggestions which have enabled us to improve its quality.
Introduction provides excessive general explanations about antibiotic alternatives, but the explanations about the availability and state of the art of different butyrate forms and justification of the hypothesis is scarce. Is there any study focusing the efficacy of butyrate forms in broilers? The study proposes to explore the effect in different variables (performance, carcass quality, meat composition, footpad dermatitis, dry matter content of feces), which are the antecedents and/or mechanisms that could justify the hypothesis.
The introduction was rewritten L73-81
Line115.- please, give information about the butyrate equivalent content in each treatment. Were they equal?
The basis for introducing the preparations to the diets was the content of butyric acid in the individual products (L). Butyrate concentration was equal in all groups, around 780 g / t of feed. The amount of the used feed additive in group 4 was 3.4 kg / t, not 4 kg / t. The relevant correction has been made. L132
Table 2, Table 3, and Table 4 should be simplified considering the different dietary periods (starter 1, starter 2, Grower 1 and 2, and finisher diets) or even reducing the number of periods, instead of a week presentation. Table 5 is repeating values shown in other Tables. Merge the information. Please consider the possibility of moving from 4 Tables to only 1 for performance results.
The relevant corrections have been made. The number of tables were reduced. Now is only 1 table for performance results.
Merge Table 6 and 7 , after deleting values from Week 6 (average information can be described in the body of the text.
Merge Table 8 and 9
The relevant corrections has been made. The number of tables were reduced.
Table 10 contain very interesting information, but it depends on prices which are continuously evolving (see for example, the extraordinary change on the prices from February 22 to April 22). This information doesn´t deserve to be published in detail.
We agree, but feed prices changed in last year every week, in fact. From this point of view is very difficult to present general economic efficiency of different forms of butyric acid. It was necessary to show this data in some period and from our perspective, the time of preparing an article it was good time to do it.
Discussion should be clearly improved. In the present version, the discussion is organized to discuss all variables sequentially with the main objective of finding previous reports that are or not coincident with these results. At his respect, discussion is not ambitious and loss the opportunity to identify likely general mechanisms that can explain the results, even considering the discussion of results on a more transversal way. As there is not a clear hypothesis, discussion doesn´t attend a main objective to identify if butyrate exert effects on turkeys, and if these effects depends on the additive forms. When differences are observed derived from the form, a proper discussion is required.
In fact, but available articles about of efficiency of different forms of BA in turkeys nutrition are scarce. From this point of view we decided to focus on general information.
The discussion was rewritten. General mechanism, that explain results can be found L.321-328, L339, L345-347, L361-366
Conclusions should not be a summary of results. For example, it is not evident if coating of glycerides promotes differences from sodium butyrate.
The conclusion was rewritten. L382,L386

Round 2
Reviewer 2 Report
Authors have made an effort to give response to some points, specially those that help to simplify the description of Results in tables.
However, a minor effort has been done in the Introduction, Discussion and Conclusion sections (see suggestion from the first review)
Author Response
The introduction was rewritten, information about antibiotic were reduced. The disccusion was rewritten, mechanism of butyric acid action was added. The relevant corrections has been made in section "Conslusions".

Round 3
Reviewer 2 Report
Done, thank you